# Improving the Interfacial Microstructure and Properties of Al/Mg Bimetal by a Novel Mo Coating Combined with Ultrasonic Field

**DOI:** 10.3390/ma18174005

**Published:** 2025-08-27

**Authors:** Jiaze Hu, Xiuru Fan, Haoheng Du, Guangyu Li, Xiaoqiong Wang, Xing Kang, Qiantong Zeng

**Affiliations:** 1School of Mechanical Engineering and Automation, Dalian Polytechnic University, Dalian 116034, China; dpuhjz@163.com (J.H.); haohengdu@163.com (H.D.); 2School of Materials Science and Engineering, Dalian University of Technology, Dalian 116024, China; xqw@mail.dlut.edu.cn (X.W.); xingkangx@163.com (X.K.); zqt_185@163.com (Q.Z.); 3Ningbo Institute of Dalian University of Technology, Ningbo 315000, China

**Keywords:** Al/Mg bimetal, microstructure, mechanical properties, Mo coating, compound casting, ultrasonic field

## Abstract

To enhance the interfacial performance of Al/Mg bimetal, this study introduced a novel Mo coating and employed an ultrasonic field (UF) to regulate the interfacial microstructure. In the absence of both a Mo coating and ultrasonic treatment (referred to as the untreated specimen), the interfacial region was primarily composed of Al-Mg intermetallic compounds (Al-Mg IMCs), Al-Mg eutectic structures (ES), and Mg_2_Si phases, with an average interfacial layer thickness of approximately 1623 μm. Upon application of the Mo coating, the formation of both Al-Mg phases and Mg_2_Si phases was completely inhibited. The interfacial zone was predominantly characterized by the Mo solid solution (Mo SS) and oxide, with the average thickness significantly reduced to about 28 μm. Upon applying the UF to the Mo-coated specimen, the interfacial composition remained similar to that of the untreated specimen, except for Mo SS, with the interfacial thickness increasing to 561 μm. Shear strength tests indicated that the application of the Mo coating alone resulted in a decrease in bonding strength compared to the untreated specimen. However, subsequent ultrasonic treatment significantly improved the interfacial shear strength to 54.7 MPa, representing a 60.9% increase relative to the untreated specimen. This improvement is primarily attributed to the Mo coating and UF synergistically suppressing the formation of brittle Al-Mg IMCs and reducing oxide inclusions at the interface. Thus, the simultaneous application of Mo coatings and ultrasonic fields is required to enhance the properties of Al/Mg bimetals.

## 1. Introduction

Magnesium alloys (Mg alloys) and aluminum alloys (Al alloys) are the two lightest metallic materials used in engineering applications [1,2,3,4]. Aluminum alloys offer advantages such as good corrosion resistance, excellent plasticity, low processing costs, and moderate strength [5], while magnesium alloys (with a density of two-thirds that of aluminum alloys) exhibit a superior specific strength, specific stiffness, thermal conductivity, shock absorption, and electromagnetic shielding properties [6]. However, magnesium alloys suffer from poor corrosion resistance, low plasticity, and high costs—deficiencies that aluminum alloys can precisely compensate for. Combining these two alloys, namely forming the Al/Mg bimetal, thus enables mutual complementation, full utilization of their respective advantages, and a balance between extreme lightweighting and structural performance, rendering them promising for applications in the automotive, low-altitude economy, aerospace, and electronic 3C fields [7].

However, Al/Mg bimetals face critical challenges, such as the difficulty in removing surface oxide films, mismatches in physical properties between Al and Mg, and the formation of brittle Al-Mg IMCs at the interface [8]. These issues result in the poor performance of bimetallic components, limiting their practical application and development. Consequently, controlling the interfacial structure of Al/Mg bimetals to enhance their properties remains a research hotspot and challenge in this field.

To improve the performance of Al/Mg bimetals, various strategies have been employed, including adjusting process parameters [9,10,11], removing surface oxide films [12,13,14], introducing intermediate layers [15,16,17,18,19,20,21,22,23,24,25,26,27,28,29,30,31,32,33,34,35,36,37,38,39,40,41,42,43,44,45,46,47,48,49,50,51,52], alloying [53,54,55,56,57,58,59], heat treatment [60], applying external fields [61,62,63,64,65,66,67,68,69,70], thermal modification [71], tailoring surface morphologies [72], and so on. Among these, introducing an intermediate layer is the most widely used and effective method, as it can both eliminate oxide films and regulate interfacial products.

Intermediate layers for Al/Mg bimetals are generally categorized into three types: (1) low-melting-point pure metals (e.g., Ga [15], Zn [16,17,18,19,20,21,22], and Sn [23,24]); (2) high-melting-point pure metals (e.g., Ni [25,26,27,28,29,30,31], Fe [32], Ti [33], Ag [34,35], Ce [36], Mn [37], and Zr [38]); and (3) alloys or non-metallic materials (e.g., Al-Mg eutectic alloys [39], Al-Ni [40], Ag-Cu-Zn [41], CuNi-Ag-CuNi [42], Cu-Ni [43], Al-Zn-Ni-Zn [44], Zn-Ni-Zn [44], Sn-coated steel [45], Zn-coated steel [45], Au-coated Ni [45], SiC [46], Al-Si [47], Ni-Cr [48], and high-entropy alloy [49,50,51,52]). Different intermediate layers exhibit distinct interfacial formation mechanisms and property regulation mechanisms: low-melting-point layers typically melt during fabrication, with their interface forming via a “melting + diffusion” mechanism, and their strengthening effect arises from improved wettability and reduced Al-Mg IMCs; high-melting-point layers generally remain unmelted, with interfaces formed primarily via elemental diffusion, and their strengthening mechanism involves eliminating brittle Al-Mg IMCs and replacing them with less brittle phases; and alloy or non-metallic layers often combine the functions of the above two types.

Most intermediate layers react with Al or Mg to form IMCs, which are generally more brittle than solid solutions [73]. This is because IMCs have complex crystal structures with fewer slip systems, making dislocation movements difficult; thus, stress relief via plastic deformation is hindered, leading to brittle fractures [74]. In contrast, solid solutions retain the crystal structure of the solvent metal, with more slip systems and easier dislocation movements, enabling energy absorption through plastic deformation and significantly lower brittleness than IMCs. Consequently, selecting an interfacial layer that reacts with only Al or Mg to form solid solutions would be an effective strategy for strengthening bimetallic materials.

Molybdenum (Mo) has a high melting point (2623 °C), much higher than those of Al and Mg, and thus remains unmelted during fabrication. According to phase diagrams [75,76], Mo reacts with Mg only to form solid solutions; although Mo can form IMCs with Al, this requires relatively harsh reaction conditions [77]. Therefore, using Mo as a coating material may result in an interfacial layer dominated by solid solutions, potentially achieving interfacial strengthening. However, studies by Kiejna A. et al. have indicated that Mo has the ability to adsorb oxygen [78], which may generate oxidation inclusions and thus be detrimental to interface strengthening. Our previous study revealed [61,62] that ultrasonic fields facilitate element diffusion, grain refinement, second-phase migration, and the reduction in oxidation inclusions. Thus, applying an ultrasonic field may address some of the challenges associated with Mo coatings. However, to the best of our knowledge, no studies have yet reported the use of Mo coatings combined with the ultrasonic field for interfacial control in Al/Mg bimetals.

This study prepares Al/Mg bimetals via the lost foam composite casting process, and the effects of Mo coatings and ultrasonic fields on the microstructure and properties of Al/Mg bimetals are focused, aiming to provide a theoretical basis and practical guidance for fabricating high-performance Al/Mg bimetallic materials.

## 2. Materials and Methods

### 2.1. Experimental Materials

The experimental materials employed in this study primarily included A356 aluminum alloy, AZ91D magnesium alloy, and Mo powder. Specifically, the A356 aluminum alloy served as the solid matrix, while the AZ91D magnesium alloy functioned as the casting melt. Their chemical compositions are presented in Table 1. Spherical Mo powder, with a particle size ranging from 45 to 75 μm, was used to fabricate a Mo coating on the surface of the A356 matrix. The morphology of the Mo powder is depicted in Figure 1a.

### 2.2. Preparation Process of Mo Coating

First, a cylindrical A356 alloy with a diameter of 10 mm and height of 130 mm was cut via wire electrical discharge machining. Following degreasing, cleaning, and sandblasting, a Mo coating was deposited on its surface using a plasma flame spraying process, as shown in Figure 1b. The Mo powder used in the preparation process was spherical pure Mo powder with a particle size of 45–75 mesh. The main process parameters for plasma spraying were as follows: (1) power: 40 kW; (2) feed rate: 8 g/min; (3) gas flow rate: 30 L/min (Ar); and (4) spraying distance: 100 mm.

The morphology and composition of the Mo coating are presented in Figure 1c,d, respectively. The coating exhibited non-uniformity with an average thickness of approximately 20 μm, as illustrated in Figure 1c, which can be attributed to its small thickness and the high surface roughness of the A356 alloy. The coating was primarily composed of Mo (99.16 wt%), with minor amounts of Si (0.06 wt%) and Al (0.78 wt%), as depicted in Figure 1d. These trace elements are likely derived from the substrate, as the elemental analysis area was relatively large and the coating was not fully dense.

### 2.3. Compound Casting Process

Herein, the lost foam solid–liquid composite casting technique was employed to fabricate Al/Mg bimetals, with the principle illustrated in Figure 2a. The fabrication process was conducted as follows:

Firstly, the A356 insert (cylindrical, as specified earlier) and the foam pattern, with the dimensions of 35 × 35 × 100 mm, were assembled to form a composite pattern. Subsequently, a dedicated lost foam casting coating was then applied to the surface of the mold and dried for subsequent use. Next, AZ91D alloy was melted under a CO_2_ + SF_6_ protective atmosphere with the temperature maintained at 730 °C. Afterwards, the composite pattern was placed into a sand box, which was then filled with loose sand and vibrated to achieve compaction. A plastic film and a sprue cup were positioned on top, and a water ring vacuum pump was used to evacuate the sand box to a pressure of 0.03 MPa. Finally, the AZ91D melt (held at 730 °C) was poured into the sand box via the sprue cup; the bimetallic casting then solidified, producing the Al/Mg bimetallic component.

Additionally, lost foam composite casting under an ultrasonic field was performed in this study. All other procedures remained consistent with those described above, except that the top of the A356 insert was connected to an ultrasonic transducer, and ultrasonic vibration was applied during pouring. The ultrasonic frequency and power were set to 20 Hz and 75 W, respectively.

Accordingly, three types of specimens were prepared: Al/Mg bimetals without a Mo coating (denoted as 0Mo specimen); with a Mo coating (20Mo specimen); and with both a Mo coating and an ultrasonic application (U-20Mo specimen).

### 2.4. Characterization

The microstructure and chemical composition of the coating and bimetallic specimens were characterized using a JSM-IT800 high-resolution scanning electron microscope (HRSEM, JEOL, Tokyo, Japan) equipped with an Oxford energy-dispersive X-ray spectrometer (EDS, Oxford Instruments, Britain, UK). Shear strength measurements of the bimetallic specimens were performed using an Instron5982 universal testing machine, and the testing principle was illustrated in Figure 2b. The shear strength was calculated according to Equation (1) [8]:(1)$$S=\frac{F}{\pi dh}$$
where *S* denotes the shear strength, *F* is the peak fore, *d* is the diameter of the cylindrical A356 insert, and *h* is the height of the shear specimen. For each parameter, three replicate specimens were tested to determine the average shear strength. The Vickers hardness distribution across the interface region was measured using a HV-1000 hardness tester (Ningbo, China, Hongjian Metallographical), with an applied load of 300 g and a dwell time of 15 s. Triplicate hardness measurements were conducted at each position, and the average value was adopted.

## 3. Results

### 3.1. Microstructure

Figure 3 presents the microstructure and EDS analysis results of the interface region for 0Mo specimen. Figure 3a reveals a distinct metallurgical reaction layer between the A356 alloy and AZ91D alloy, which differs significantly from the matrix. The metallurgical reaction layer is relatively thick, with an approximate thickness of 1623 μm. Based on EDS line scanning and mapping results, the metallurgical reaction layer exhibits heterogeneity, revealing three distinct subregions: an Al-rich region (layer I) adjacent to A356, a Mg-rich region (layer III) adjacent to AZ91D, and an intermediate region (layer II) enriched in both Mg and Si, as illustrated in Figure 3b–e. Furthermore, different positions within the interface layer exhibit distinct morphological characteristics.

High-magnification observations and further compositional analyses of these subregions are presented in Figure 3f–i and Table 2. Both region A and region B show a morphology of particulate second phases distributed on a smooth matrix, as exhibited in Figure 3f,g. According to EDS analysis results, the particulate second phase is identified as Mg_2_Si, and the matrices of regions A and B are Al_3_Mg_2_ and Al_12_Mg_17_, respectively. The difference between the two regions lies in the distribution and size of Mg_2_Si. The size of Mg_2_Si in region A is highly inhomogeneous, with the fine Mg_2_Si particles being much smaller in size than those in region B. In contrast, the Mg_2_Si in region B exhibits a regular polygonal morphology with a relatively uniform size. Region D displays a typical chrysanthemum-like morphology of a eutectic structure, which is determined by EDS analysis to be an Al_12_Mg_17_ + Mg eutectic structure, as indicated in Figure 3i. Region C between regions D and B is a transition zone featuring morphological characteristics of both regions D and B, as demonstrated in Figure 3h. Moreover, continuous black strips and some black phases are observed in this region, which are confirmed by EDS analysis as oxidation inclusions. Furthermore, numerous Al_12_Mg_17_ dendrites were observed within the eutectic layer, as indicated by the yellow arrows in Figure 3a. Thus, layer I consists primarily of Al_3_Mg_2_ and Mg_2_Si (Figure 3f); layer II is dominated by Al_12_Mg_17_ and Mg_2_Si (Figure 3g); and layer III is primarily composed of an Al_12_Mg_17_ + δ-Mg eutectic structure (Figure 3i). Herein, Al_3_Mg_2_ and Al_12_Mg_17_ are collectively referred to as Al-Mg IMCs. Layers I and II can thus be designated as Al-Mg IMC layers. The Al_12_Mg_17_ + δ-Mg eutectic structure is hereinafter abbreviated as ES. Accordingly, the phase composition of the interfacial layer in the 0Mo specimen can be summarized as Al-Mg IMCs + Mg_2_Si + ES. This finding is consistent with those reported in previous studies [79,80,81].

Figure 4 presents the microstructure and EDS analysis results of the interface region of 20Mo specimen. Figure 4a reveals that the thickness of the interfacial layer is significantly reduced to approximately 28 μm after the introduction of the Mo coating. EDS results indicate that the interfacial layer is predominantly composed of the Mo element, with a relatively high oxygen content near the AZ91D side, as illustrated in Figure 4b–f. High-magnification micrographs of the interfacial region and corresponding EDS results are provided in Figure 4g,h and Table 3. Figure 4g shows that the interface layer primarily consists of a light gray region (layer I) adjacent to the A356 side and a dark region (layer II) adjacent to the AZ91D side. The EDS point analysis at location 1 reveals a composition of 95.6 at% Mo, 3.62 at% Mg, and 0.78 at% Al, thus this region is identified as a Mo solid solution (Mo SS). EDS analyses at Points 2 and 3 indicate that O is the dominant element in the dark region, confirming it to be an oxide. Additionally, the EDS analysis at Point 4 identifies the phase at this location as Al_12_Mg_17_, likely a precipitated phase within the AZ91D matrix. Figure 4g also reveals fine dark phases distributed within the Mo SS layer. To clarify their compositions, this region was further magnified for microstructural and compositional analysis, with results presented in Figure 4h and Table 3 (for Points 5–11). These analyses confirm that the dark phases within the Mo SS layer are also oxides. Overall, the interfacial layer composition of the Mo-coated bimetal is summarized as consisting of Mo SS and oxides, exhibiting a marked compositional difference compared to the 0Mo specimen.

Figure 5 presents the microstructure and EDS results of the U-20Mo specimen. Figure 5a reveals that the thickness of the interfacial layer is approximately 561 μm under ultrasonic treatment, which is significantly greater than that of the 20Mo specimen but smaller than that of the 0Mo specimen. Based on the results of EDS line scanning and mapping scanning, the interfacial layer can be roughly divided into three subregions: an Al-rich subregion (layer I) adjacent to A356, a Mg-rich subregion (layer III) adjacent to AZ91D, and a Mo-rich subregion (layer II). The Mo-rich subregion is concentrated primarily at the junction of layer I and layer III, with a small portion distributed within layer III.

To confirm the phase composition of these subregions, high-magnification observations and EDS analyses were performed, with results presented in Figure 5g–j and Table 4. EDS results for Points 1 and 2 in Figure 5g reveal that layer I is primarily composed of Al_3_Mg_2_ + Mg_2_Si. According to EDS analyses in Figure 5h,j, layer III consists mainly of Al_12_Mg_17_, Mg_2_Si, and a eutectic structure—components identical to those in the 0Mo specimen. Similarly, Al_12_Mg_17_ dendrites were observed in the eutectic region, but their quantity was lower than in the 0Mo specimen. Figure 5i and corresponding EDS results indicate that layer II is dominated by Mo SS with a small amount of oxides, bounded by Al-Mg IMCs and Mg_2_Si, on one side and the eutectic structure on the other. Overall, the phase composition of the interfacial layer in the U-20Mo specimen can be summarized as Al-Mg IMCs + Mg_2_Si + Mo SS + ES.

Image Pro Plus 6.0 (IPP) software was employed to measure the thickness of reaction layers in different specimens and calculate the proportion of each reaction layer within the entire interfacial layer. The results are presented in Table 5 and Figure 6. These results indicate that the thickness of the interfacial layer in both 20Mo and U-20Mo specimens is significantly lower than that in the 0Mo specimen. Furthermore, it is noted that the proportion of thickness of the Al-Mg IMCs layer for the U-20Mo specimen is also significantly lower than that for the 0Mo specimen.

### 3.2. Mechanical Properties

Figure 7 presents the Vickers hardness distribution across the interface for different Al/Mg bimetallic specimens. As revealed in Figure 7, the hardness of the interfacial layer is significantly higher than that of the A356 and AZ91D matrices, with variations observed among different reaction layers. Specifically, the ES layer exhibits the lowest hardness, ranging from 160 to 170 HV, while the Mo SS layer shows the highest hardness at approximately 330–390 HV. The hardness of the Al-Mg IMC layer, lower than that of the Mo SS layer, ranges from 260 to 290 HV.

Figure 8 presents the shear test curves and shear strength results of different Al/Mg bimetallic specimens. Specifically, the shear strength of the 0Mo specimen is 34.0 MPa, while that of the 20Mo specimen decreases to 26.7 MPa. When an ultrasonic field is applied, the shear strength of the U-20Mo specimen is significantly improved to 54.7 MPa, increasing by 60.9% compared to the 0Mo specimen and by 104.9% compared to the 20Mo specimen, respectively. These results indicate that merely introducing a Mo coating not only fails to enhance the shear strength but also leads to a reduction in it. In contrast, the shear strength can be effectively improved through the combined effect of the Mo coating and ultrasonic field. The underlying mechanisms accounting for this phenomenon will be discussed in the subsequent section.

Figure 9 presents the cross-sections and fracture surfaces of different bimetallic specimens. Figure 9a–c show that fractures in all bimetallic specimens occurred within the interfacial layer, indicating that the interfacial layer is the weakest region. However, the fracture initiation sites and crack propagation paths differ slightly among the specimens. Figure 9a reveals that the fracture of the 0Mo specimen initiates at the interface between the Al-Mg IMCs layer and the A356 matrix. The crack propagates along the Al-Mg IMCs layer to the junction between the Al-Mg IMCs layer and the ES layer, with a small portion extending into the ES layer. Figure 9b shows that the fracture of the 20Mo specimen occurs at the interface between the Mo SS and AZ91D, propagating along this interface until complete fracture. For the U-20Mo specimen, the fracture initiates at the Al-Mg IMCs layer, with the crack propagating to the ES layer. Furthermore, Figure 9d–f indicate that the fracture surfaces contain numerous cleavage steps, river patterns, and cleavage fans, exhibiting typical cleavage fracture features. Thus, all three specimens undergo brittle fracture.

## 4. Discussion

### 4.1. Interfacial Formation Mechanism

From the aforementioned microstructural analyses, it is evident that the microstructural characteristics and phase compositions of the interfacial layer in Al/Mg bimetals vary under different conditions, arising from differences in their interface formation mechanisms. Herein, the formation mechanisms of the interfacial layer are discussed in detail, as illustrated in Figure 10.

In the absence of a Mo coating, a dense oxide film is present on the surface of the A356 insert, as shown in Figure 10a. When the high-temperature AZ91D melt vaporizes the foam pattern and comes into contact with the A356 insert, several concurrent phenomena occur, as indicated in Figure 10(a1): (1) Since the pouring temperature exceeds the melting point of the A356 insert, the surface of the A356 insert melts into a liquid state; (2) owing to the relatively low surface temperature of the A356 insert, significant compositional undercooling arises at the interface when the high-temperature AZ91D melt initially contacts the A356 surface, leading to the preferential formation of dendrites—this explains the numerous Al_12_Mg_17_ dendrites observed in the eutectic layer of Figure 3a; (3) the dense oxide film is partially disrupted by the scouring action of the flowing molten metal, resulting in the discontinuous oxide film observed in Figure 3h; (4) some decomposition products of the foam pattern are not completely expelled from the mold cavity and remain in the molten metal; and (5) driven by concentration gradients, Mg elements diffuse toward the A356 side, while Al and Si elements diffuse toward the AZ91D side. Based on the Al-Mg binary phase diagram [82], Al_3_Mg_2_, Al_12_Mg_17_, and Al_12_Mg_17_ + δ-Mg eutectic structures form sequentially from the Al side to the Mg side, corresponding to compositional variations. Additionally, according to the Mg-Si binary phase diagram [82], Mg and Si can only form the Mg_2_Si phase, which accounts for the numerous Mg_2_Si phases observed in the interfacial layer. Thus, the solidified interface structure, as shown in Figure 10(a2), comprises Al-Mg IMCs, Mg_2_Si, ES, and some oxide inclusions. Finally, the interface formation mechanism of the 0Mo specimen can be summarized as a ‘melting + diffusion’ process.

When the surface of the A356 insert is coated with a 20 μm Mo coating, the Mo coating effectively inhibits the oxidation of the insert’s surface, eliminating the formation of a dense oxide film, as exhibited in Figure 10b. Given that the melting point of Mo (2623 °C) significantly exceeds the temperature of the molten metal (730 °C), the Mo coating remains intact during the casting process. Consequently, only the interdiffusion of Al, Si, Mg, and Mo elements occurs at the interface, as presented in Figure 10(b1). Simultaneously, decomposition products from the foam pattern are partially entrapped in the molten metal, while Mo exhibits a strong affinity for oxygen [78], leading to the formation of an O-rich layer adjacent to the Mo coating. This promotes the diffusion of O toward the Mo coating. According to the Al-Mo binary phase diagram (Figure 11a) [75], Al and Mo form a solid solution when the Al content is below 8 at%. Similarly, the Mg-Mo binary phase diagram (Figure 11b) indicates that Mg and Mo also form a solid solution [76]. Therefore, after solidification, the interfacial region of the 20Mo specimen consists solely of Mo SS and oxides, as demonstrated in Figure 10(b2). Finally, the interface formation mechanism of 20Mo specimen can be summarized as element diffusion.

The interface formation mechanism of U-20Mo is illustrated in Figure 10(c–c2). Ultrasonic treatment exerts two primary effects: vibration and acoustic cavitation [61,62]. Owing to the non-uniform thickness of the Mo coating, vibration-induced detachment occurs at thinner regions, disrupting the integrity of the coating. Consequently, the interfacial reactions observed in both the 0Mo and 20Mo specimens occur concurrently, as shown in Figure 10(c1). The resultant interfacial layer comprises Al-Mg IMCs, Mg_2_Si, Mo SS, and ES. However, the presence of the Mo coating moderates the Al-Mg metallurgical reaction, resulting in a thinner interfacial layer. Additionally, ultrasonic vibration fragments the dendrites, reducing the density of Al_12_Mg_17_ dendrites at the interface and eliminating secondary dendrite arms (Figure 5a). Acoustic cavitation refers to the formation, growth, oscillation, and violent collapse of microbubbles in a liquid medium subjected to ultrasonic waves (mechanical waves > 20 kHz) [65]. In this study, cavitation-induced bubble dynamics facilitate the floating of oxide inclusions, minimizing interfacial defects, as depicted in Figure 10(c1). Consequently, the U-20Mo specimen exhibits negligible oxide inclusions at the interface. Finally, the interface formation mechanism of the U-20Mo specimen can also be summarized as melting and diffusion.

### 4.2. Interfacial Enhancement Mechanism

The fracture modes of different bimetallic specimens vary due to disparities in their interfacial layers, as shown in Figure 12. Fracture toughness, a critical parameter reflecting a material’s resistance to crack propagation and structural integrity, exhibits an inverse relationship with fracture susceptibility [48]. To analyze the fracture toughness of distinct reaction layers, SEM images of hardness indentations were acquired, as exhibited in Figure 13. Figure 13a–c reveal that the Al-Mg IMCs layer generates multiple long and straight cracks under external loading, whereas the ES layer exhibits fewer and shorter cracks. This indicates a lower fracture toughness in the Al-Mg IMCs layer compared to the ES layer. Consistent with the findings of G.Y. Li et al. [53], Al_3_Mg_2_ and Al_12_Mg_17_ are brittle phases, while ES is a ductile structure. Moreover, Al_3_Mg_2_ exhibits a lower fracture toughness than Al_12_Mg_17_, aligning with the observed crack propagation behavior. Thus, for the 0Mo specimen under shear loading, cracks initiate in the Al_3_Mg_2_ layer and propagate along it. Upon encountering grain boundaries or precipitates, secondary cracks form and extend toward oxide inclusions, which exhibit the lowest fracture toughness. Consequently, fractures occur predominantly within the Al-Mg IMCs layer and oxide inclusions, as indicated in Figure 12a, manifesting as a typical brittle fracture.

The SEM image of the Mo SS layer in Figure 13d shows only short cracks at hardness indentations, indicating a superior toughness compared to the Al-Mg IMCs layer. However, the presence of a porous oxide layer in the 20Mo specimen leads to fracture initiation at this position, as demonstrated in Figure 12b, resulting in a lower shear strength than the 0Mo specimen.

Under ultrasonic treatment (U-20Mo specimen), although cracks still initiate in the Al-Mg IMCs layer, the elimination of oxide inclusions enables crack propagation into the ES and Mo SS layers, as exhibited in Figure 12c, which possess a higher toughness. This transition significantly enhances shear strength. Additionally, the thickness and proportion of the brittle Al-Mg IMCs have both decreased according to Figure 6. In summary, the enhanced shear strength of the U-20Mo specimen arises from the synergistic effects of eliminating oxide inclusion defects and reducing the proportion of brittle phases at the interface.

## 5. Conclusions

This study primarily investigates the effects of a Mo coating and ultrasonic field on the microstructure and properties of the Al/Mg bimetallic system. The main conclusions are as follows:(1)In the absence of a Mo coating, the interfacial layer between Al and Mg has a thickness of approximately 1623 μm, predominantly composed of Al-Mg IMCs, Mg_2_Si, and ES. With the introduction of a 20 μm Mo coating, the high melting point of the Mo coating inhibits the Al-Mg interfacial reaction, leading to the complete elimination of Al-Mg phases and Mg_2_Si in the interfacial layer. Instead, the interfacial layer is dominated by Mo SS and oxide inclusions, with its thickness reduced to 28 μm. Under ultrasonic field, the Mo coating is partially fractured at localized regions, causing phases identical to those in the 0Mo specimen with a Mo SS layer. Consequently, the interfacial layer thickness decreases to 561 μm, accompanied by a reduced proportion of the Al-Mg IMC layer.(2)The microhardness of the interfacial layer is significantly higher than that of the matrix, with slight variations among different reaction sublayers. Specifically, the Mo SS layer exhibits the highest hardness (330–390 HV), while the eutectic layer shows the lowest hardness (160–170 HV). Shear strength measurements indicate that the 20Mo specimen has a lower shear strength than the 0Mo specimen. Under ultrasonic treatment, the shear strength increases significantly to 54.7 MPa, representing a 60.9% improvement compared to the 0Mo specimen. Fracture analysis reveals that all three specimens exhibit brittle fracture characteristics.(3)The interfacial formation and fracture mechanisms vary among different specimens. For the 0Mo specimen, the interfacial formation mechanism involves melting and diffusion. In the presence of a high-melting-point Mo coating, interface formation is dominated by elemental diffusion. Under ultrasonic treatment, due to the ultrasonic vibration-induced disruption of the Mo coating’s integrity, its interface formation mechanism reverts to melting and diffusion. The enhanced shear strength of the U-20Mo specimen compared to the 0Mo specimen is attributed to the elimination of oxide inclusion defects and a reduced proportion of Al-Mg brittle phases. Thus, the combination of the Mo coatings and ultrasound field is required to achieve the strengthening effect on Al/Mg bimetals.

## Figures and Tables

**Figure 1 materials-18-04005-f001:**
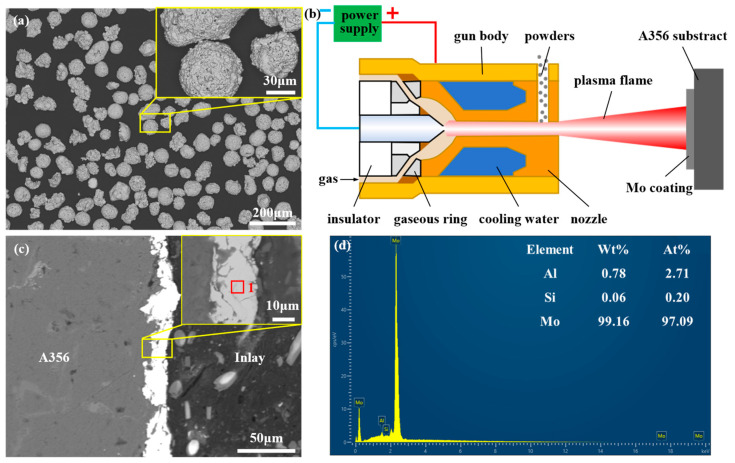
The raw materials and preparation process of the Mo coating: (**a**) morphology of Mo powder; (**b**) schematic diagram of plasma flame spraying; (**c**) morphology of Mo coating; and (**d**) composition of the Mo coating corresponding to point 1 in (**c**).

**Figure 2 materials-18-04005-f002:**
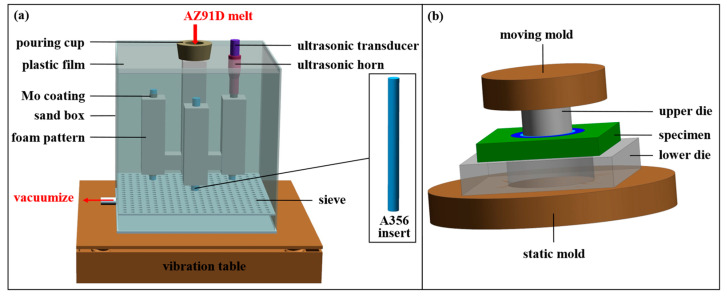
Schematic diagram of lost foam casting (**a**) and schematic diagram of shear test (**b**).

**Figure 3 materials-18-04005-f003:**
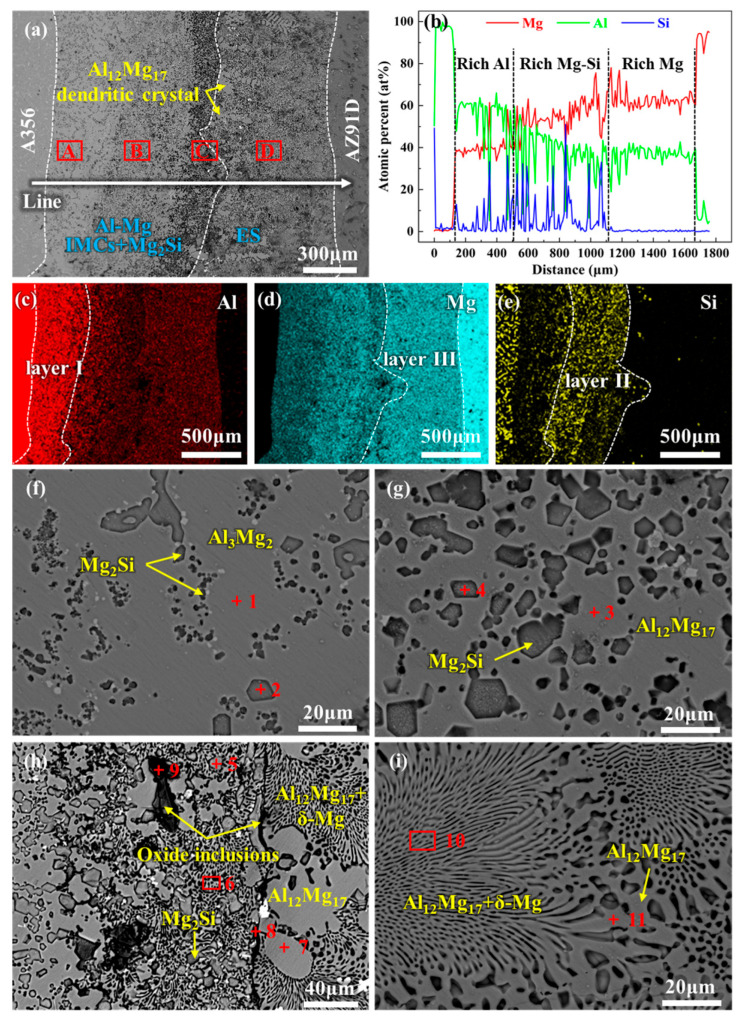
SEM image and EDS analysis results of the interface region of 0Mo specimen: (**a**) SEM image of the whole interface region; (**b**) line scanning corresponding to the position indicated by the white straight line in (**a**); (**c**–**e**) are the map scanning results of Al, Mg, and Si, respectively, corresponding to the whole region of (**a**); and (**f**–**i**) are the high-magnification SEM images of the A, B, C, and D regions in (**a**), respectively.

**Figure 4 materials-18-04005-f004:**
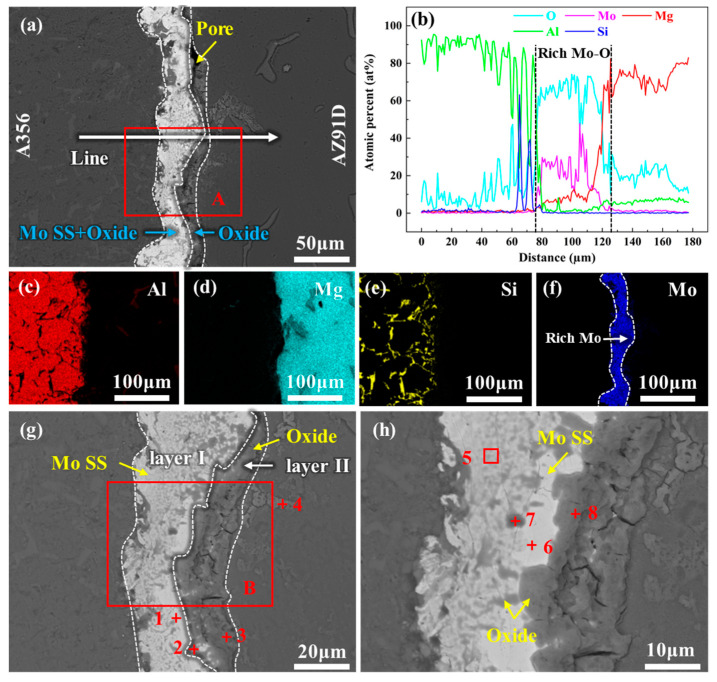
SEM image and EDS analysis results of the interface region of 20Mo specimen: (**a**) SEM image of the whole interface region; (**b**) line scanning corresponding to the position indicated by the white straight line in (**a**); (**c**–**f**) are the map scanning results of Al, Mg, Si, and Mo, respectively, corresponding to the whole region of (**a**); (**g**) high-magnification SEM image of A region in (**a**); and (**h**) high-magnification SEM image of B region in (**g**).

**Figure 5 materials-18-04005-f005:**
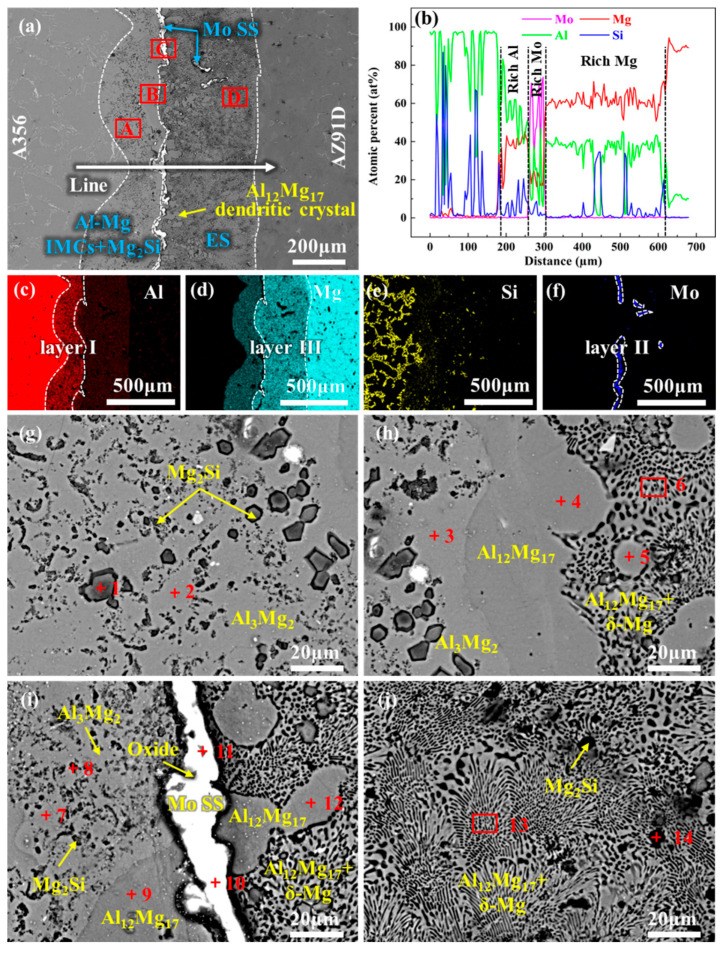
SEM image and EDS analysis results of the interface region of U-20Mo specimen: (**a**) SEM image of the whole interface region; (**b**) line scanning corresponding to the position indicated by the white straight line in (**a**); (**c**–**f**) are the map scanning results of Al, Mg, Si, and Mo, respectively, corresponding to the whole region of (**a**); and (**g**–**j**) are the high-magnification SEM images of the A, B, C, and D regions in (**a**), respectively.

**Figure 6 materials-18-04005-f006:**
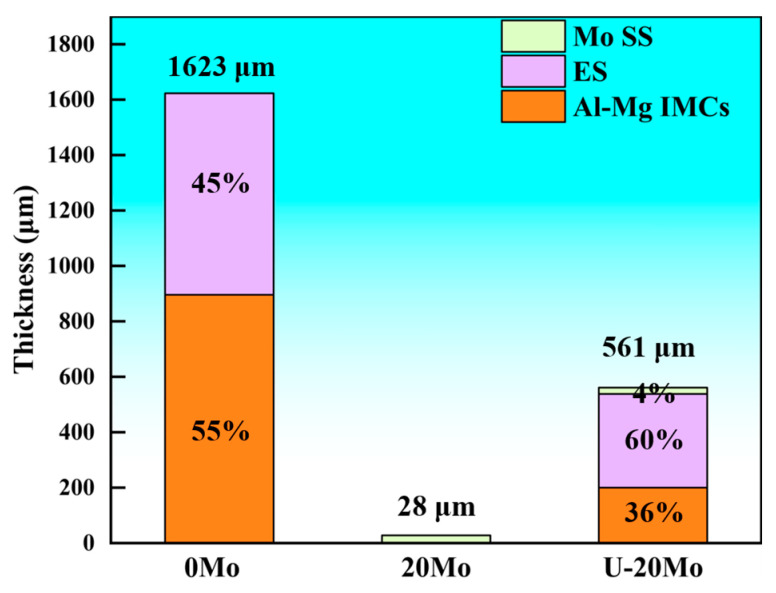
Proportion of different reaction layers.

**Figure 7 materials-18-04005-f007:**
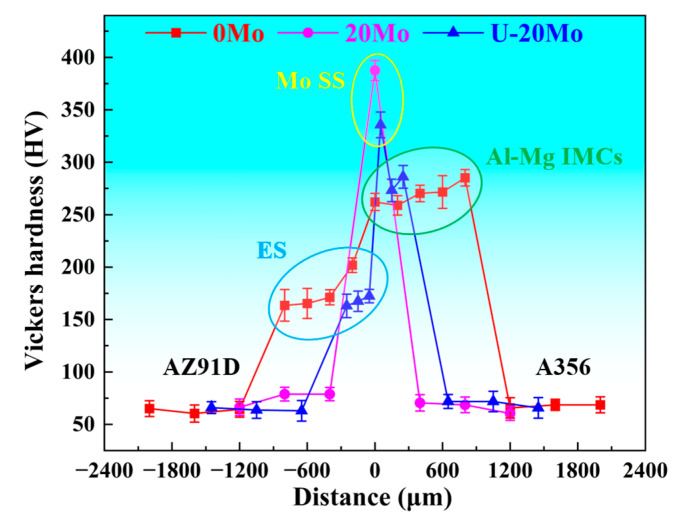
Vickers hardness distribution of the interfacial region.

**Figure 8 materials-18-04005-f008:**
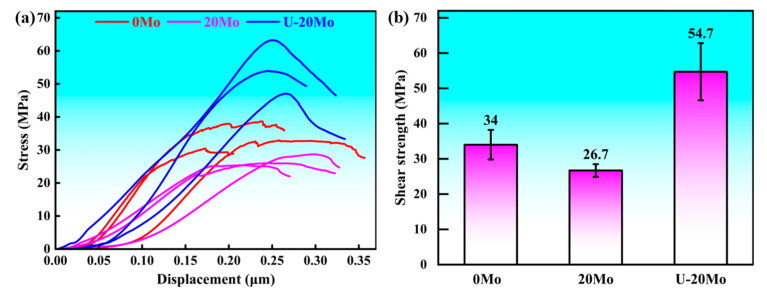
Shear test results of different Al/Mg bimetallic specimens: (**a**) Displacement–Strength curve; (**b**) shear strength.

**Figure 9 materials-18-04005-f009:**
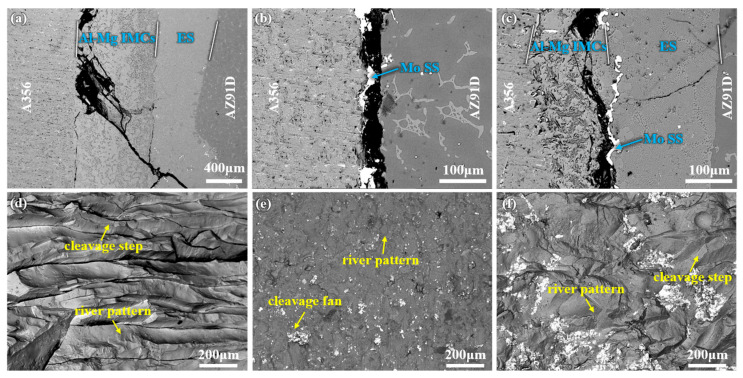
The fracture morphologies of different Al/Mg bimetallic specimens: (**a**–**c**) are the SEM images of the longitudinal sections of the 0Mo, 20Mo, and U-20Mo shear fracture specimens, respectively; (**d**–**f**) are the SEM images of the fracture surfaces of the 0Mo, 20Mo, and U-20Mo shear fracture specimens, respectively.

**Figure 10 materials-18-04005-f010:**
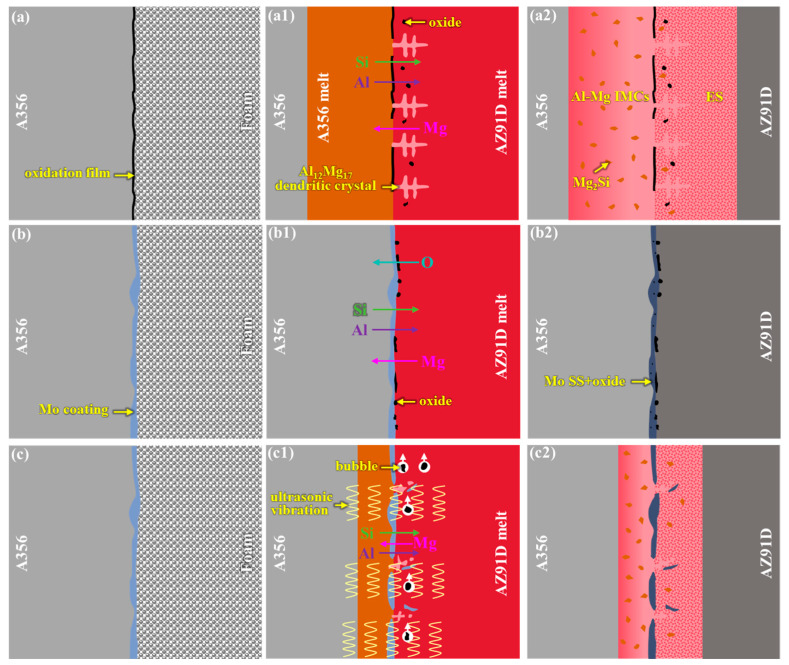
Schematic diagram of the interface formation mechanism of different bimetallic specimens: (**a**), (**a1**,**a2**) 0Mo specimen; (**b**), (**b1,b2**) 20Mo specimen; and (**c**), (**c1,c2**) U-20Mo specimen.

**Figure 11 materials-18-04005-f011:**
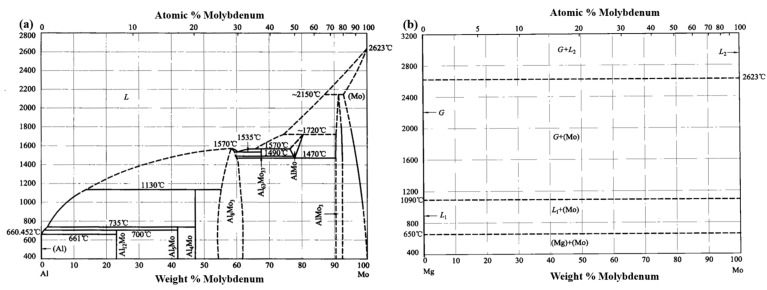
Binary phase diagram: (**a**) Al-Mo binary phase diagram [75]; (**b**) Mg-Mo binary phase diagram [76].

**Figure 12 materials-18-04005-f012:**
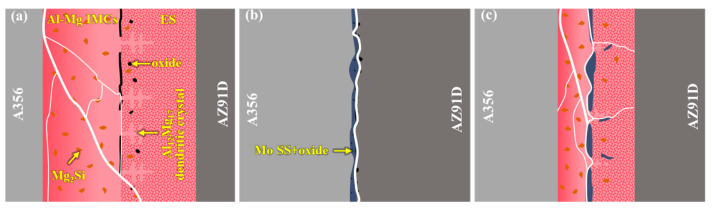
Fracture mechanisms of different bimetallic specimens: (**a**) 0Mo specimen; (**b**) 20Mo specimen; and (**c**) U-20Mo specimen.

**Figure 13 materials-18-04005-f013:**
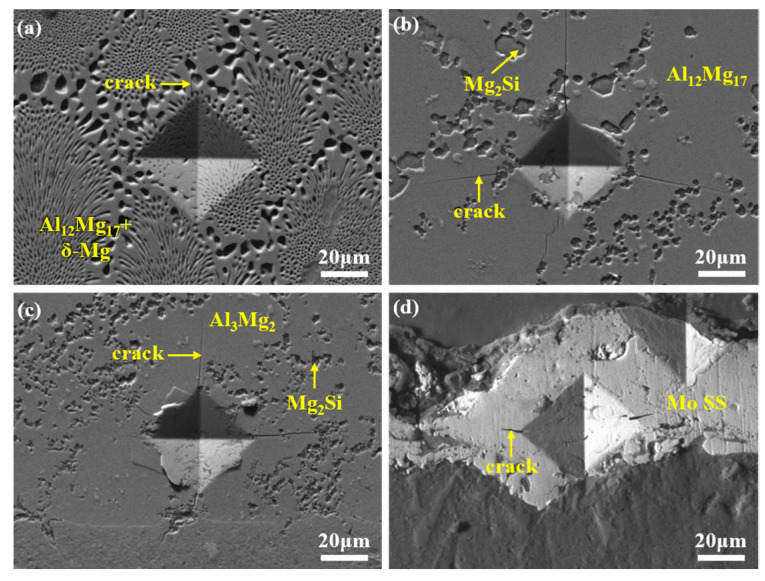
SEM images of hardness points in different regions: (**a**) ES layer; (**b**) Al_12_Mg_17_ + Mg_2_Si layer; (**c**) Al_3_Mg_2_ + Mg_2_Si layer; and (**d**) Mo SS layer.

**Table 1 materials-18-04005-t001:** The compositions of A356 and AZ91D.

Element	Mass Fraction (%)						
	Mn	Zn	Si	Ti	Fe	Mg	Al
A356	-	-	6.81	0.017	0.205	0.439	Bal.
AZ91D	0.62	0.23	-	-	-	Bal.	9.08

**Table 2 materials-18-04005-t002:** EDS results of different points signed in Figure 3.

Position	Element (at%)				Phase
	Al	Mg	Si	O	
1	63.56	36.44	-	-	Al_3_Mg_2_
2	-	58.48	41.52	-	Mg_2_Si
3	42.20	57.80	-	-	Al_12_Mg_17_
4	-	62.08	37.92	-	Mg_2_Si
5	-	57.15	42.85	-	Mg_2_Si
6	10.28	89.72	-	-	ES
7	40.38	59.62	-	-	Al_12_Mg_17_
8	12.71	23.15	-	64.14	Oxide
9	11.24	33.22	8.76	46.78	Oxide
10	11.35	88.65	-	-	ES
11	37.29	62.71	-	-	Al_12_Mg_17_

**Table 3 materials-18-04005-t003:** EDS results of different points signed in Figure 4.

Position	Element (at%)					Phase
	Al	Mg	Si	Mo	O	
1	0.78	3.62	-	95.60	-	Mo SS
2	2.84	14.57	-	15.18	67.4	Oxide
3	3.71	32.06	-	15.71	48.51	Oxide
4	44.72	55.28	-	-	-	Al_12_Mg_17_
5	1.98	8.39	-	17.70	71.92	Oxide
6	0.78	4.31	-	94.91	-	Mo SS
7	4.01	13.01	-	21.56	61.42	Oxide
8	2.44	23.45	-	8.95	65.16	Oxide

**Table 4 materials-18-04005-t004:** EDS results of different points signed in Figure 5.

Position	Element (at%)					Phase
	Al	Mg	Si	Mo	O	
1	-	35.71	64.29	-	-	Mg_2_Si
2	38.01	61.99	-	-	-	Al_3_Mg_2_
3	38.95	61.05	-	-	-	Al_3_Mg_2_
4	47.85	52.15	-	-	-	Al_12_Mg_17_
5	40.13	59.87	-	-	-	Al_12_Mg_17_
6	14.99	85.01	-	-	-	ES
7	59.59	40.41	-	-	-	Al_3_Mg_2_
8	-	36.66	63.34	-	-	Mg_2_Si
9	48.40	51.60	-	-	-	Al_12_Mg_17_
10	4.38	24.46	1.23	69.92		Mo SS
11	11.40	12.28	-	13.53	62.79	Oxide
12	38.98	61.02	-	-	-	Al_12_Mg_17_
13	11.29	88.71	-	-	-	ES
14	-	33.87	66.13	-	-	Mg_2_Si

**Table 5 materials-18-04005-t005:** Different thicknesses of reaction layers.

Specimen	Thickness (μm)			
	Al-Mg IMCs	ES	Mo SS	Overall Interface Layer
0Mo	896	727	0	1623
20Mo	0	0	28	28
U-20Mo	200	338	23	561

## Data Availability

The original contributions presented in this study are included in the article. Further inquiries can be directed to the corresponding authors.

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
