# Peer review of "Improving the Interfacial Microstructure and Properties of Al/Mg Bimetal by a Novel Mo Coating Combined with Ultrasonic Field"

_materials, 2025, doi:10.3390/ma18174005_

Round 1

Reviewer 1 Report

Comments and Suggestions for Authors

The article is devoted to the creation of Al/Mg bimetallic material. To improve the properties of this material, the authors proposed to use novel Mo coating combined with ultrasonic field.
When reviewing the article, I got a good impression of the study presented by the authors. In my opinion, this article can be published in the scientific journal Materials without additional correction.
I will note some important points.
1. The title of the article "Improving the interfacial microstructure and properties of Al/Mg bimetal by a novel Mo coating combined with ultrasonic field" fully reflects the essence of the study and allows the reader to understand what this work is about.
2. The abstract reflects the main results achieved in the study and indicates their numerical characteristics. In this form, the Abstract allows the reader to understand the need to study the full version of the article.
3. The analysis of previously completed studies is presented competently and reasonably. The authors analyzed 77 studies. The analysis includes both studies from the last 5 years and earlier ones. Based on the analysis results, the authors formulated the objective of the study: This study prepares Al/Mg bimetals via the lost foam composite casting process, employing plasma-sprayed Mo coatings on A356 substrates as a strengthening approach. The primary focus is on investigating the effects of Mo coatings and ultrasonic fields on the microstructure and properties of Al/Mg bimetals, aiming to provide a theoretical basis and practical guidance for fabricating high-performance Al/Mg bimetallic materials
4. The materials and methods of the study are described in great detail in Section 2. The graphic material is of very high quality and significantly facilitates understanding the features of the studies conducted.
5. The graphic material in Section 3. Results is presented competently and has a high-quality design. The markers and inscriptions are clear and of good size. This significantly facilitates understanding the results obtained.
6. In Section 4. Discussion, the authors explain the results obtained in great detail. They accompany their reasoning with diagrams and explanatory drawings.
7. The conclusions of the research reflect the essence and the results achieved. However, in my opinion, a summary point could be added here, which would reflect recommendations for the use of Mo coating and ultrasonic exposure modes. The authors have similar recommendations in the Abstract. 
8. I am curious to know why the authors chose the plasma flame spraying method of coating application. Is it possible to apply Mo coating in another way? What are the advantages of this method? The coating turns out to be quite loose and uneven, which the authors themselves talked about in this work.

Reviewer 2 Report

Comments and Suggestions for Authors

This is an interesting work, very well structured and prepared. Then, I have some minor issues that need to be solved before publication in materials described as follows:

+ Which equipment and details were used to obtain the SEM-EDS figures?

+ Please, include a figure of the cylindrical A356 sample.

+ Please change the formula to the equation, and include the reference where this equation is located 1.

+ Why about the surface roughness of the coating?

+ How many times was the Vickers hardness measured by condition? Also, which equipment was used? 

+ Any standard was considered to the Vickers measurements?

+ Any DRX pattern? Should be interesting.

+  What is the preferred growth direction of this coating?

+ Please talk about the morphology in a deep way, considering the microstructure presented in Figure 3. It looks different depending on the condition.

+ Please use more technical terms,,, by examen change images by micrographs.

Reviewer 3 Report

Comments and Suggestions for Authors

The manuscript evaluates the impact of a Mo coating and ultrasonic field on the mechanical properties of the A356/AZ91D bimetallic composite. The Mo layer inhibits the formation of the brittle intermetallic layer, while the ultrasonic field minimizes oxide formation at the interface. The results are are clearly presented, and the discussion introduces interesting findings. However, to enhance the clarity of the study, the following points require attention:

  1. There are several typographical errors throughout the manuscript.
  2. The nomenclature used to refer to the Al/Mg bimetal without Mo coating is inconsistent (e.g., “0Mo” vs. “Mo0”). Standardize this throughout the text.
  3. In the Introduction, cite the reasons for using the ultrasonic field to prepare Al-Mg bimetals.
  4. Some experimental details are missing: the Mo powder composition, the plasma spraying parameters, and the dimensions of the foam pattern.
  5. Figure 4b: In the Mo–O-rich layer, the atomic percentage of oxygen consistently exceeds that of Mo. Is this a genuine compositional feature or a limitation of the line scanning technique used?
  6. Figures 6 and 8: The label of the first bar is wrong.
  7. In the displacement-strength curve, the U-20Mo sample exhibits significant variation between specimens. What could be the cause of this variability? Does the process ensure adequate repeatability, or are there factors contributing to inconsistency?
  8. P. 8, lines 226-227: "Al₁₂Mg₁₇ dendrites were observed in the eutectic region, but 226 their quantity was lower than in the Mo0 specimen" and p. 15, lines 390-391: "ultrasonic fragmentation of Al₁₂Mg₁₇ dendrites, effectively refining 390 the grain structure". The claim that improved shear strength is due to grain refinement lacks strong supporting evidence and appears speculative. Unless further analyses are provided, this statement should be removed, and related discussions and conclusions revised accordingly.
  9. Since the use of Mo as an interlayer is a novel contribution of this study, it would be beneficial to compare its effectiveness with that of other intermediate layers reported in the literature.

Round 2

Reviewer 3 Report

Comments and Suggestions for Authors

I would like to thank the authors for addressing my points. No further comments.